# Observation of domain wall bimerons in chiral magnets

Tomoki Nagase [1✉], Yeong-Gi So[2], Hayata Yasui[2], Takafumi Ishida[3,4], Hiroyuki K. Yoshida[5], Yukio Tanaka[4], Koh Saitoh[3,4], Nobuyuki Ikarashi[1,6], Yuki Kawaguchi [4], Makoto Kuwahara[3,4✉] & Masahiro Nagao [1,6✉]

Topological defects embedded in or combined with domain walls have been proposed in various systems, some of which are referred to as domain wall skyrmions or domain wall bimerons. However, the experimental observation of such topological defects remains an ongoing challenge. Here, using Lorentz transmission electron microscopy, we report the experimental discovery of domain wall bimerons in chiral magnet Co-Zn-Mn(110) thin films. By applying a magnetic field, multidomain structures develop, and simultaneously, chained or isolated bimerons arise as the localized state between the domains with the opposite in-plane components of net magnetization. The multidomain formation is attributed to magnetic anisotropy and dipolar interaction, and domain wall bimerons are stabilized by the Dzyaloshinskii-Moriya interaction. In addition, micromagnetic simulations show that domain wall bimerons appear for a wide range of conditions in chiral magnets with cubic magnetic anisotropy. Our results promote further study in various fields of physics.

[1] Department of Electronics, Graduate School of Engineering, Nagoya University, Nagoya, Japan. [2] Department of Materials Science, Graduate School of Engineering Science, Akita University, Akita, Japan. [3] Advanced Measurement Technology Center, Institute of Materials and Systems for Sustainability, Nagoya University, Nagoya, Japan. [4] Department of Applied Physics, Graduate School of Engineering, Nagoya University, Nagoya, Japan. [5] Department of Physics, Faculty of Science, Hokkaido University, Sapporo, Japan. [6] Center for Integrated Research of Future Electronics, Institute of Materials and Systems for Sustainability, Nagoya University, Nagoya, Japan. ✉email: nagase.tomoki@k.nagoya-u.jp; kuwahara@imass.nagoya-u.ac.jp; nagao.masahiro@imass.nagoya-u.ac.jp

Topological defects arise when symmetry is spontaneously broken. They have long been studied because of playing key roles in understanding of diverse systems ranging from cosmological length scales to condensed matter[1,2]. A domain wall (DW) is a type of topological defect, which is the boundary between domains with different signs of order parameters in a conventional sense. DWs are interesting not only for understanding nature but also for applications. For example, in ferromagnets, multiferroics, and magnetic topological insulators where an order parameter is the magnetization, the manipulation of magnetic DWs could potentially enable the realization of devices such as high-performance memory devices[3], energy conversion devices[4], and quantum information processing[5].

A skyrmion is another type of topological defect which can occur in two or three dimensions and was first introduced in the field of nuclear physics. Skyrmions are now widespread and realized in various systems[6–8]. Among them, although two-dimensional magnetic skyrmions have only a short history since the discovery in a chiral magnet[9], they are gaining attention as a platform for studying emergent electromagnetic fields and related physical properties owing to the real-space topology of their spin textures[10]. In particular, their electric current-driven motion at ultralow current density is expected to be exploited in modern efficient spintronic devices.

Meanwhile, topological defects embedded in or combined with DWs have been proposed in various systems so far. Malozemoff and Slonczewski have discussed the effect of such topological defects on magnetic DW mobility, where the topological defects are Bloch lines that exist as line defects in two-dimensional DWs[11]. Usually, Bloch lines are formed by accidental excitation. Subsequently, Fal'ko and Iordanskii have first predicted pairs of stable Bloch lines in DWs in quantum Hall ferromagnet at the filling factor $\nu = 1$ (ref. [12]). Such topological defects are then called DW skyrmions. It is noted, however, that skyrmions are mathematically classified by the second homotopy group $\pi_2(S^2)$, whereas merons are classified by the relative second homotopy group $\pi_2(S^2,S^1)$ (refs. [13,14]). The difference is the boundary condition on a circle surrounding the object: The magnetization direction around a skyrmion is fixed, meanwhile that around a meron winds with nonzero winding number $\pi_1(S^1)$. Thus, the topological defect that can exist inside a DW is not a skyrmion but a meron, and we refer it to as not a DW skyrmion but a DW bimeron. Ever since then, DW bimerons have been independently proposed using various theoretical models in $\nu = 2$ quantum Hall ferromagnet[15], liquid crystals[16], and similar topological defects in various systems[17–23]. In addition, more recently, magnetic DW bimerons have also been predicted[24,25]. Although DW bimerons have attracted considerable interest because of involving a wide-ranging field of physics described above, no clear observation has been achieved so far[26,27].

Real-space imaging offers direct evidence for the presence of DW bimerons. We show the simulated Lorentz transmission electron microscopy (LTEM) images of magnetic skyrmions and conventional DWs as an example. DW bimerons would be identified as an LTEM image where the contrasts at DWs have similar to that of skyrmions. In magnets, LTEM can image both DWs and skyrmions, as illustrated in Fig. 1. The imaging principle is as follows. The incident electron beam is deflected by the Lorentz force and simultaneously its phase of the wavefunction is shifted by the in-plane component of the magnetization when passing through a magnetic thin film, and then the defocused interference image is seen on the screen. Qualitatively, the bright (dark) contrast of the LTEM images can be understood as the convergent (divergent) of the electron beam. The bright and dark contrasts reverse for the underfocused and overfocused images. In the case of Bloch DWs (Fig. 1a), the contrast pairs of the bright

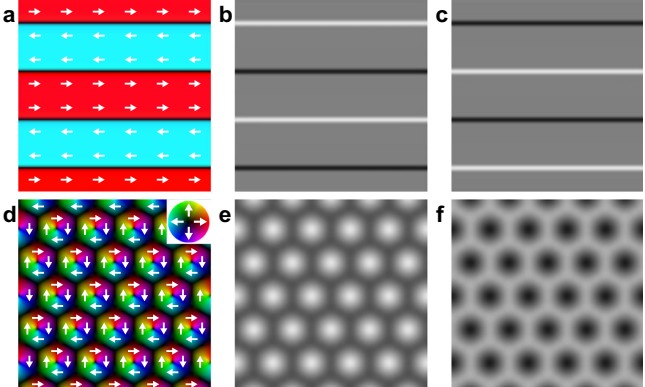

**Fig. 1 Schematic of DWs and skyrmions, and the corresponding LTEM images. a** Magnetic structure of DWs in ferromagnets. **b**, **c** Corresponding LTEM images of DWs for underfocused (**b**) and overfocused (**c**) conditions. **d** Magnetic structure of skyrmions. **e**, **f** Corresponding LTEM images of skyrmions for underfocused (**e**) and overfocused (**f**) conditions.

and dark lines are observed (Fig. 1b, c). In the case of Bloch skyrmions (Fig. 1d), either bright or dark dot contrasts are observed (Fig. 1e, f).

In this study, using LTEM, we report the direct observation of DW bimerons in cubic β-Mn-type chiral magnet Co–Zn–Mn(110) thin films. As magnetic fields are applied perpendicular to the planes, we observe the development of multidomain structures and two types of DWs, where one is a chain of DW bimerons and the other is a conventional DW. Bimeron chains and conventional DWs alternatively appear. The formation of the multidomains and paired DW structures are attributed to the combination of magnetic anisotropy, dipolar interaction, and the Dzyaloshinskii–Moriya interaction (DMI). In addition, the numerical simulations incorporating the well-known cubic magnetic anisotropy provide that DW bimerons can appear for a wide range of conditions in chiral magnet thin films.

## Results

**Imaging magnetic structures with LTEM**. We investigate Co$_{8.5}$Zn$_{7.5}$Mn$_4$(110) thin film with a thickness of $t$ ~50 nm (see Methods). This material has the Curie temperature of 345 K and helical spin period of ~145 nm and β-Mn-type materials have larger magnetic anisotropy than B20-type materials hosting skyrmions and the magnetic easy axes are along the <100> directions[28]. Figure 2 shows the magnetization process with increasing magnetic fields perpendicularly downward to the plane at 330 K (for further LTEM images, see Supplementary Figs. 1, 2). A black line contrast identifying as a DW exists at a zero magnetic field (Fig. 2a). With increasing magnetic field, the additional black lines (conventional DWs) develop along the <110> direction, whereas the chains of the bright elliptical dots also develop (Fig. 2b). The individual bright dots are similar to the skyrmion contrasts. The conventional DWs and chains are always paired. By further increasing magnetic fields, the conventional DWs and chains develop further and show an increase in number up to 170 mT (Fig. 2c, d), and then turn to a decrease in number (Fig. 2e). Finally, the conventional DWs and chains almost disappear (Fig. 2f).

In DW bimerons, bimerons play the role of the domain boundaries, in contrast to the conventional magnetic skyrmions in the uniform conical or field-polarized background[9,29]. Our LTEM images (Fig. 2) imply that the chains of the bright elliptical dots are DW bimerons. However, the LTEM images alone are not clear: it is necessary to clarify whether there are domains with different magnetization directions on both sides of the chains. We

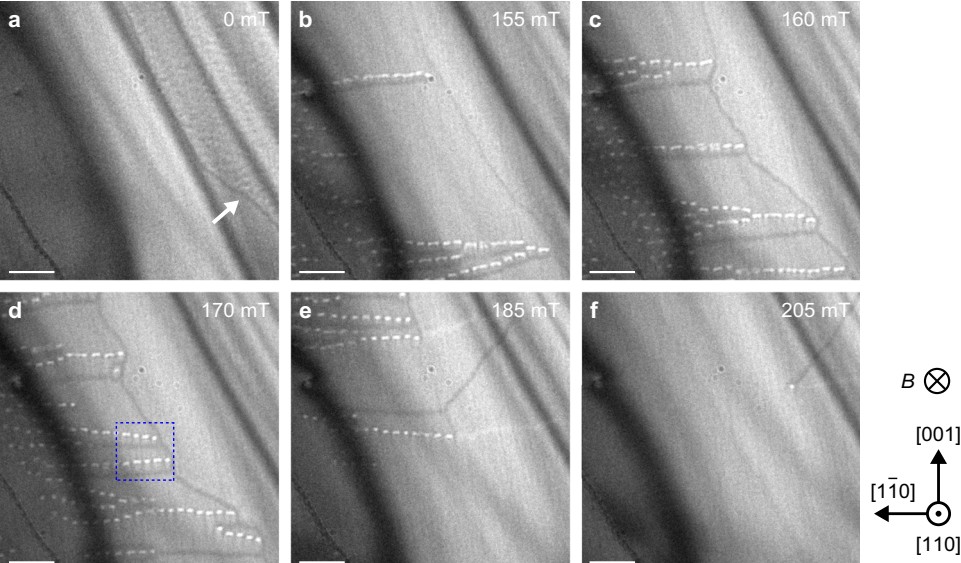

**Fig. 2 Development of magnetic structures in the Co$_{8.5}$Zn$_{7.5}$Mn$_4$(110) thin film with $t$ ~50 nm.** A series of underfocused LTEM images (defocus value $\Delta f = -2$ mm) obtained at 330 K and 0 mT (**a**), 155 mT (**b**), 160 mT (**c**), 170 mT (**d**), 185 mT (**e**), and 205 mT (**f**). The magnetic field was applied perpendicular to the thin film plane. The scale bars are 1 μm. The strong dark line contrasts (the so-called bend contours) seen in the upper left to lower middle of each figure appear at locations where the Bragg condition is locally satisfied due to sample bending. Further details are displayed in Supplementary Figs. 1, 2. The blue dotted box in **d** corresponds to Fig. 3a.

analyse the LTEM images to reveal the magnetic distributions. Figure 3a, b are the underfocused and overfocused LTEM images, respectively. We have used the transport of intensity equation based on Fig. 3a, b (see Methods). Figure 3c shows the in-plane magnetic flux density map. We note that Fig. 3c almost shows magnetization texture because the effects of leakage flux and demagnetizing field are small enough (see Methods). In addition to the Bloch magnetic texture of each vortex and DW, the domains with the opposite magnetizations along <110> directions are definitely formed on both sides of the vortex chains. Thus, our data provide direct evidence of magnetic DW bimerons. (The termination of the chains such as in the upper right in Fig. 3a, b, c may not have a topological charge of 1/2 and may have that of zero.) Furthermore, at 324 K, we found isolated bimerons bound to DWs, as presented in Fig. 3f. (The enlarged image within the blue dotted box in Fig. 3f is shown in Fig. 3g.) Since the number of observed bimerons decreases away from the Curie temperature, the formation is likely to be assisted by thermal fluctuations, as in the case of skyrmions. Further details of the temperature variation of magnetic structures are shown in Supplementary Fig. 3. In the following, bimerons playing a role of and bound to DWs are referred to as DW bimerons for convenience.

**Formation mechanism.** The formation mechanism of observed DW bimerons is different from that of theoretical predictions[24,25] and our previous study[30]. Our achievement is attributed to the combination of the thin film thickness with $t$ ~50 nm, magnetic anisotropy, dipolar interaction, and DMI. Our previous experiment on the same material Co$_{8.5}$Zn$_{7.5}$Mn$_4$(110) thin films with $t$ ~100 nm reported that an applied magnetic field perpendicular to the plane induces a transition from a stripe phase to a smectic liquid-crystalline phase of skyrmions without any multidomains, due to magnetic anisotropy and partially modulated spin structures[30]. In the smectic phase, skyrmions exist within the uniform conical domain or field-polarized background[30] as with conventional skyrmions. On the other hand, in the present study, in-plane helical spin orders with propagation vector perpendicular to the plane stabilize at a zero magnetic field, which has

often been observed in the thin films of β-Mn-type Co–Zn–Mn (refs. [31,32]), and the sample thickness (~50 nm) is about a third of the helical spin period (~145 nm). As a result, propagating in-plane helical spin orders are truncated before going around once in the present thin film. Truncated in-plane helical spin orders lead to a finite in-plane net magnetization, which is the magnetic structure of the domains. Figure 2a indeed shows two domains having different initial phases of spin at the sample surface (bottom) which are separated by the DW, due to dipolar inter-action and wedge-shaped sample geometry effect. In conventional ferromagnet thin films, when a magnetic field is applied per-pendicular to the plane, the volume fraction of domains only changes and the number of domains does not increase, but, in our thin film, multidomains accompanied by DWs and DW bimerons develop (Fig. 2b, c, d). Considering the combined effects of magnetic anisotropy and dipolar interaction can explain the formation and development of multidomains. In our thin film, one magnetic easy axis is parallel to the in-plane direction, and the other two are at an angle of 45° to the in-plane direction. The latter two have the components parallel to both the <110> directions within the plane and the direction of the applied magnetic field. By a magnetic field is applied perpendicular to the plane, magnetostatic energy and Zeeman energy mainly compete. In particular, due to magnetic induction, the increase in mag-netostatic energy becomes quite large in the present thin film, whose thickness is ~50 nm, which is much thinner than that of the previous study[30]. Here, to reduce magnetostatic energy by dipolar interaction, the initial phase of spins on the sample sur-face (bottom) chooses between two equivalent magnetic easy axes in the <110> directions, resulting in truncated in-plane conical spin orders with a finite in-plane net magnetization, as shown in Fig. 3d, e where the arrows depict perpendicularly averaged magnetization to the plane. Therefore, multidomains composed of truncated conical spin orders with different initial phases of spins on the sample surface (bottom) develop. It is noted that LTEM detects the average value in the film thickness direction of the in-plane component of magnetic flux, therefore, DWs between truncated conical domains are imaged similarly as with

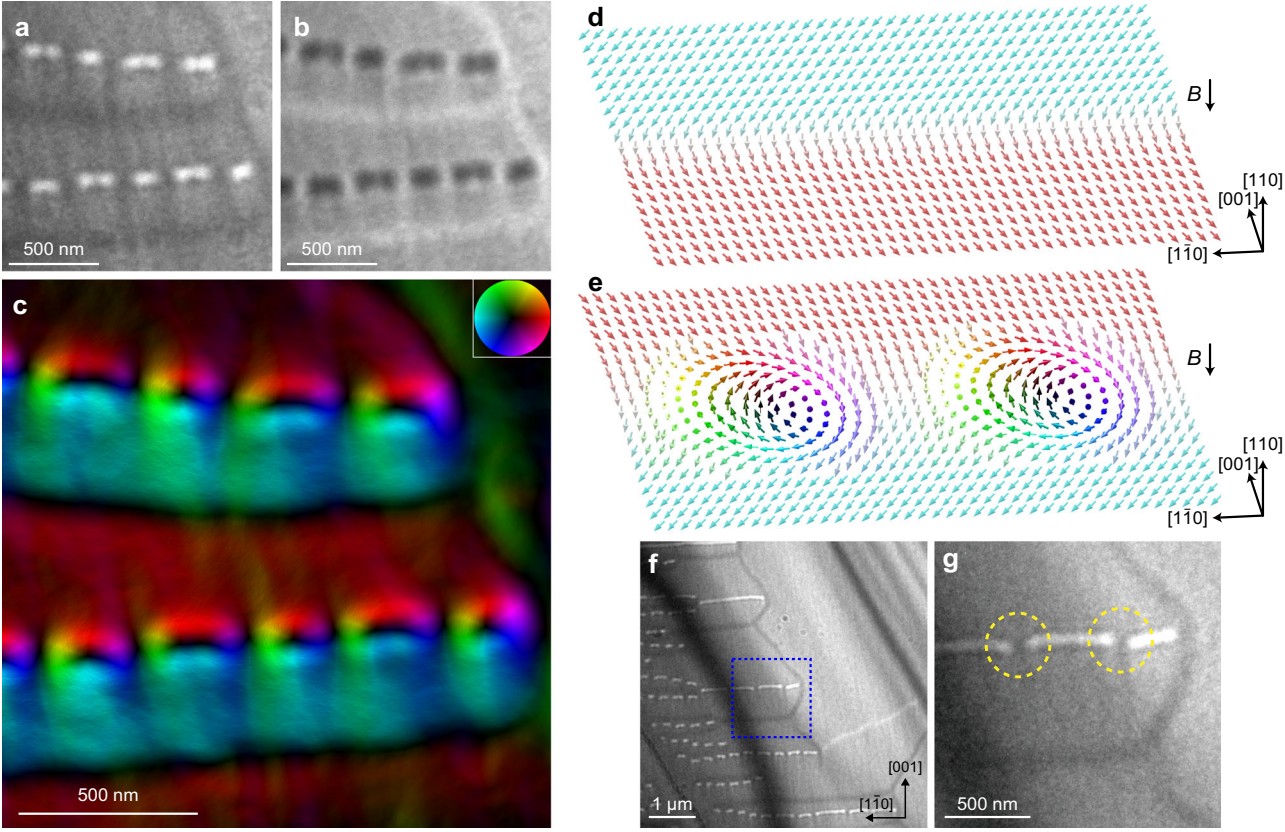

**Fig. 3 Magnetic structure of DW bimerons and isolated DW bimerons. a, b** Underfocused ($\Delta f = -1.5\,\mathrm{mm}$) (**a**) and overfocused ($\Delta f = +1.5\,\mathrm{mm}$) (**b**) LTEM images, enlarged view of the blue dotted box in Fig. 2d. The scale bars are 500 nm. **c** Magnetic flux density (magnetization) map obtained using the transport of intensity equation analysis based on **a** and **b**. The scale bar is 500 nm. The color wheel (inset) represents the orientation and magnitude of the in-plane component of magnetic flux density. The in-plane magnetizations on both sides of the bimeron chains have the opposite directions along the in-plane <110> directions, verifying the existence of DW bimerons. **d, e** Schematic illustration of DW (**d**) and DW bimerons (**e**). The arrows depict perpendicularly averaged magnetization to the plane. In the domains, average magnetization are oriented in the <100> directions of the magnetic easy axes. **f** Underfocused LTEM image with $\Delta f = -2\,\mathrm{mm}$ obtained at 324 K and 190 mT. **g** Enlarged view of the blue dotted box in **f**.

DWs in conventional ferromagnets. Of particular importance to form DW bimerons is the DMI that restricts the magnetic chirality of DWs. By applying a magnetic field perpendicular downwardly to the plane, DWs with the downward spin rotation are stable (Fig. 3d), while those with upward spin rotation are unstable, and instead, bimerons are stabilized at the boundaries between the domains (Fig. 3e). The formation of DW bimerons, which differ dramatically from that of smectic skyrmions[30] at different film thicknesses, is due to the combination as above described, which has been not predicted.

**Micromagnetic simulations**. To verify the above surmise, the micromagnetic simulation seems to be an effective method. In fact, despite using the simple micromagnetic model expressed only by terms describing the ferromagnetic exchange interaction, DMI, Zeeman, and dipole interaction energies, it can well reproduce the microscopic internal spin textures of skyrmions and their array in chiral magnets such as B20-type FeGe (ref. [33]). However, it is known that it is difficult to completely reproduce the magnetic structure of β-Mn-type Co–Zn–Mn by the micromagnetic simulation[30]. The reason may be due to the atomic-scale complexities of the Co–Zn–Mn. This material has the two crystallographic sites randomly occupied by the two or three elements and the coexistence of the ferromagnetic coupling between Co–Co, Co–Mn, and the antiferromagnetic coupling between Mn–Mn (ref. [34]). Due to these specific material

properties, it is regrettably difficult for the Co–Zn–Mn to fully reproduce the magnetic structures, and therefore in future, more complex simulations should be developed.

Here, apart from the complex Co–Zn–Mn, we numerically propose that DW bimerons appear for a wide range of conditions in cubic chiral magnets that have magnetic anisotropy. We have performed the micromagnetic simulation using the simple model described above with the addition of the cubic magnetocrystalline anisotropy[28] (see Methods). The term of the magnetocrystalline anisotropy is described by $K_c\{(\boldsymbol{m}\cdot\boldsymbol{C}_1)^2(\boldsymbol{m}\cdot\boldsymbol{C}_2)^2 + (\boldsymbol{m}\cdot\boldsymbol{C}_2)^2(\boldsymbol{m}\cdot\boldsymbol{C}_3)^2 + (\boldsymbol{m}\cdot\boldsymbol{C}_3)^2(\boldsymbol{m}\cdot\boldsymbol{C}_1)^2\}$, where $K_c$ is an anisotropy constant, $\boldsymbol{m}$ is the unit magnetization vector, and $\boldsymbol{C}_1, \boldsymbol{C}_2, \boldsymbol{C}_3$ are the unit vectors along the <100> directions. $K_c$ in our simulations is the same in order of magnitude as that of other works[28,35]. The magnetic easy axes are the <100> directions ($K_c > 0$) (ref. [28]) and the magnetic field ($B$) is applied perpendicularly downward to the (110) thin film plane with $t = 50$ nm. The simulated magnetic structures are summarized in Fig. 4a where the colors indicate the orientation and magnitude of perpendicularly averaged in-plane components of magnetization to the plane. At zero and small $K_c$ (lower left area), the conventional triangular skyrmion lattice is stabilized. At large $K_c$, on the other hand, triangular skyrmion lattice is unstable, and instead, domains with the opposite net magnetizations composed of truncated conical spin orders (see also Supplementary Fig. 4) are stabilized into strips along the <110> directions. Accordingly, the Bloch DWs and the vortex chains are

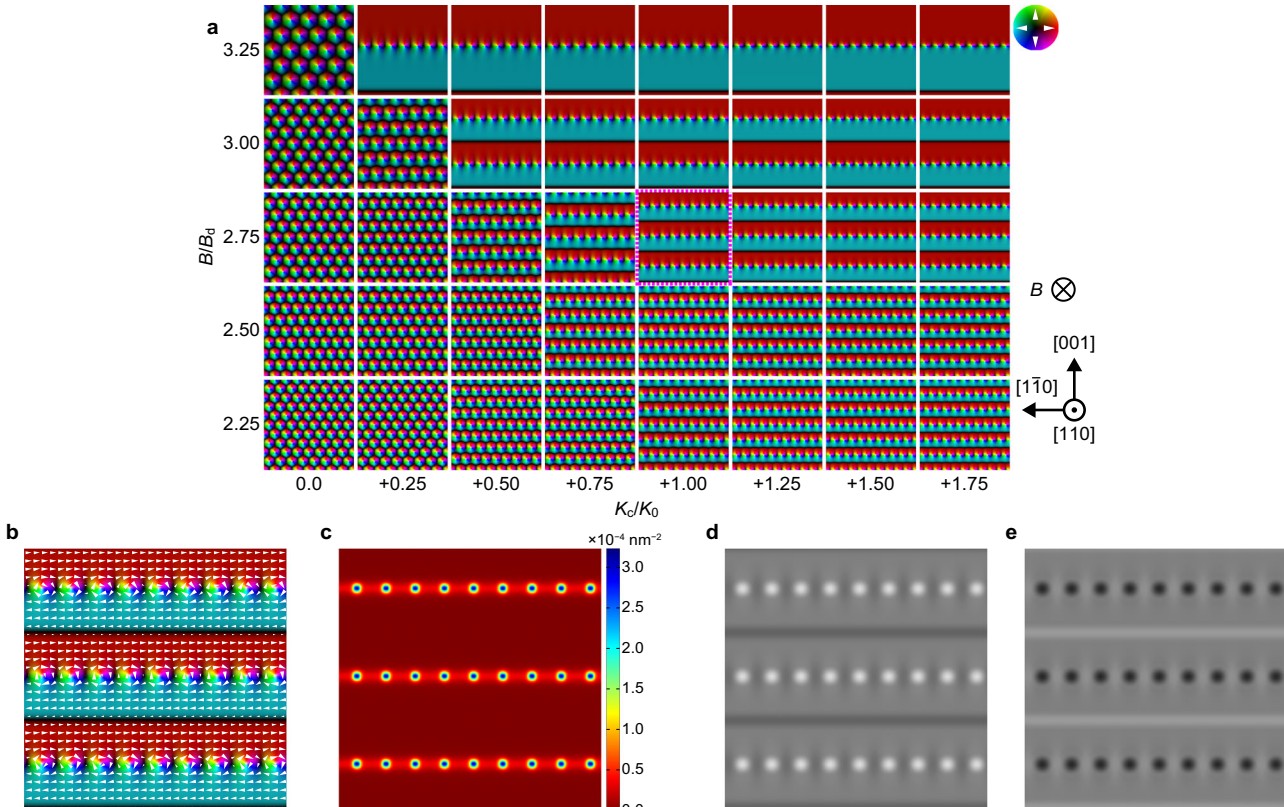

**Fig. 4 Micromagnetic and LTEM image simulations of DW bimerons in a cubic chiral magnet (110) thin film with $t = 50$ nm. a** Simulated results of magnetization distributions for various cubic magnetocrystalline anisotropy ($K_c$) and magnetic fields ($B$). Parameters are as follows: saturation magnetization $M_{sat} = 180$ kA/m, exchange stiffness constant $A = 3.83$ pJ/m and DMI constant $D = 0.332$ mJ/m$^2$. $K_0 = D^2/4A$ and $B_d = D^2/2M_{sat}A$ are normalization constants. The magnetic easy axes are the <100> directions ($K_c > 0$). The size of each image is 1.6 × 1.6 μm$^2$. The color wheel (upper right panel) represents the orientation and magnitude of perpendicularly averaged in-plane components of truncated in-plane conical magnetization to the plane. The white and black indicate $+z$ and $-z$ directions, respectively. **b** Enlarged view of magnetic structure corresponding to magenta dotted box in **a**. **c** Calculated TCD map for **b**. **d, e** Simulated underfocused ($\Delta f = -1.5$ mm) (**d**) and overfocused ($\Delta f = +1.5$ mm) (**e**) LTEM images for **b**. Further details are shown in Supplementary Fig. 4.

formed alternately as the domain boundaries. To confirm that the vortex chains are DW bimerons, in Fig. 4c, we profile the topological charge density (TCD) for the simulation result (Fig. 4b). In the TCD map, there are peaks at each vortex, while no peaks at the Bloch DWs. The integration value of the TCD, which is the sum of half the regions of the two adjacent vortices, is +1, indicating the bimeron spin texture[24]. Furthermore, the simulated underfocused and overfocused LTEM images for the magnetic distributions (see Methods) are almost in agreement with the experimental LTEM images (Fig. 3a, b). There is a little difference between the simulation and experimental results due to the complexities of the Co–Zn–Mn: the circular and elliptical vortices in the simulation and experiment, respectively. Incidentally, our simulation also exhibits DW bimerons along the <100> directions when the easy axes are the <111> directions (Supplementary Fig. 5). In our experiment, DW bimerons along the <100> directions actually appear in Co$_7$Zn$_{11}$Mn$_2$(110) thin film (Supplementary Fig. 6) where magnetic anisotropy may vary with the composition in the Co–Zn–Mn.

The formation conditions of DW bimerons are summarized as follows: (I) a cubic chiral magnet has large magnetocrystalline anisotropy, (II) the thin film has the (110) plane, (III) the thin film thickness is not equal to the integral multiple of the helical period for having a finite net magnetization of domains, and (IV) dipolar interaction is fully effective to form multidomains. Our micromagnetic simulation suggests that DW bimerons appear for

a wide range of conditions in chiral magnets that meet the above formation conditions, for instance, a skyrmion-material Cu$_2$O-SeO$_3$ with large magnetocrystalline anisotropy[36,37]. Cu$_2$O-SeO$_3$(110) thin films when the above conditions are satisfied is expected that bimerons are formed as DWs in the vicinity of the Curie temperature of $T_C \sim 60$ K due to thermal fluctuations and bound to DWs in temperatures away from $T_C$ as well as the present study of the Co–Zn–Mn(110) thin films.

## Discussion

In the field of physics related to topology such as field theory and Bose–Einstein condensates, bimerons and merons/antimerons have long been paid much attention. For example, the bound states of fractional vortices are discussed in terms of quark confinement, where baryons and mesons are considered to be bimerons and pairs of meron–antimeron, respectively (refs. [38,39]). In magnetism, merons/antimerons and bimerons have currently attracted considerable interest in spintronic applications utilizing topological properties as well as fundamental physics, and their lattice forms and isolations have proposed to be formed by in-plane magnetic anisotropy[40–42]. In a chiral magnet thin film supposedly with in-plane magnetic anisotropy, a recent experiment showed a square lattice form of merons/antimerons, which stimulate further investigation of emergent electromagnetic properties as well as skyrmions[32]. On the other hand, in our chiral magnet thin films, we show chained and isolated bimerons

playing a role of and bound to DWs, respectively, which are realized by not only in-plane magnetic anisotropy component but also the combination of DMI, out-of-plane magnetic anisotropy, dipolar interaction, and Zeeman effect. These unique bimeron states and their formation mechanism have not been predicted in previous studies and our findings provide a new platform of discussion for topology in condensed matter as well as for field theory and Bose–Einstein condensates. It is noted that an isolated bimeron bound to a DW is a defect within a defect. Such kinds of defects have been discussed so far in various systems[21]. In magnets, Bloch lines are randomly formed defects due to dipolar interaction. However, in chiral magnets that we studied here, isolated bimerons are selectively formed in one side of DWs only due to the above combination. They are found for the first time in this study, and in an application, expected to be driven along DWs which could avoid a phenomenon similar to the Hall effect that impedes device realization[24].

Our LTEM images have demonstrated the experimental observation of magnetic DW bimerons, pairing up with the conventional DWs in chiral magnet Co–Zn–Mn(110) thin films. In addition, our micromagnetic simulations propose that DW bimerons can appear for a wide range of conditions in cubic chiral magnet (110) thin films with large magnetocrystalline anisotropy. Our results provide a guide to realize DW bimerons and promote further study in diverse fields of physics.

## Methods

**Sample preparation and LTEM observations.** The bulk polycrystalline samples of β-Mn-type Co–Zn–Mn were prepared from highly pure Co (99.995%), Zn (99.99%), and Mn (99.99%) by a conventional melting process. The constituents of the alloys sealed in an evacuated quartz tube were heated at 1273 K for 12 h, then slowly cooled to 1198 K at a cooling rate of 1 K/h, and kept at the temperature for 72 h, followed by water-quenching. The magnetization measurement was performed using the superconducting quantum interference device. The grains are large enough to manipulate a piece of the bulk samples as a single crystal. The thin films for the LTEM observations were prepared by the following procedures. Pieces of the bulk samples were mechanically cut out parallel to the (110) plane and then was thinned by Ar-ion milling method. The composition and the thickness of the thin film were measured by the energy dispersive X-ray analysis and the electron energy-loss spectroscopy, respectively. Observations of magnetic structures were performed using the Fresnel mode of LTEM (JEM2100F, JEOL) with the specimen-heating double-tilting holder. The displayed specimen temperature is calibrated and the LTEM images are obtained after taking enough time to stabilize the temperature. To avoid the artificially-elongated magnetic contrasts, the astigmatism was corrected using magnification calibration diffraction grating replica. The value of the magnetic field was changed by controlling the current of the objective lens. The locations of the LTEM images were calibrated using the scratches and contaminations on the specimen as marks. Figure 3c is obtained from the LTEM images using QPt software package (HREM Co.) which is based on the transport of intensity equation[43].

**Micromagnetic simulation.** Micromagnetic simulations were performed using Mumax3 (ref. [44]). Magnetic structures are simulated by full energy minimization using a conjugate gradient method for the thin plate of a chiral magnet with a size of $1600 \times 1600 \times 50$ nm³ on a $384 \times 384 \times 12$ mesh. The continuum energy function ($\varepsilon$) which contains the ferromagnetic exchange interaction, the DMI, the demagnetizing field energy, the Zeeman energy, and the cubic magnetocrystalline anisotropy energy is described by

$$\varepsilon = A\left\{\left(\frac{\partial \boldsymbol{m}}{\partial x}\right)^2 + \left(\frac{\partial \boldsymbol{m}}{\partial y}\right)^2 + \left(\frac{\partial \boldsymbol{m}}{\partial z}\right)^2\right\} + D\boldsymbol{m}\cdot(\nabla \boldsymbol{m}) - \frac{1}{2}M_{\text{sat}}\boldsymbol{m}\cdot\boldsymbol{B}_{\text{demag}} - M_{\text{sat}}\boldsymbol{m}\cdot\boldsymbol{B}$$
$$+ K_c\left\{(\boldsymbol{m}\cdot\boldsymbol{C}_1)^2(\boldsymbol{m}\cdot\boldsymbol{C}_2)^2 + (\boldsymbol{m}\cdot\boldsymbol{C}_2)^2(\boldsymbol{m}\cdot\boldsymbol{C}_3)^2 + (\boldsymbol{m}\cdot\boldsymbol{C}_3)^2(\boldsymbol{m}\cdot\boldsymbol{C}_1)^2\right\}. \quad (1)$$

$A$, $D$, $M_{\text{sat}}$, $\boldsymbol{B}_{\text{demag}}$, $\boldsymbol{B}$, and $K_c$ are the exchange stiffness constant, the micromagnetic constant of the DMI, the saturation magnetization, the demagnetizing field, the magnetic field, and the cubic anisotropy constant, respectively. $C_i = C_{ix}e_x + C_{iy}\boldsymbol{e}_y + C_{iz}\boldsymbol{e}_z$ ($i = 1, 2, 3$) is a unit vector along the <100> directions corresponding to (110) thin film.

$$\begin{cases} \boldsymbol{C}_1 = \boldsymbol{e}_y \\ \boldsymbol{C}_2 = \frac{1}{\sqrt{2}}\boldsymbol{e}_x + \frac{1}{\sqrt{2}}\boldsymbol{e}_z \\ \boldsymbol{C}_3 = \frac{1}{\sqrt{2}}\boldsymbol{e}_x - \frac{1}{\sqrt{2}}\boldsymbol{e}_z \end{cases} \quad (2)$$

Here, $\boldsymbol{e}_i$ is a fundamental unit vector in the Cartesian coordinate system ($i = x$, $y$,

$z$). Periodic boundary conditions are applied in the $x$- and $y$-directions and Neumann boundary conditions[44,45]

$$\left.\frac{\partial \boldsymbol{m}}{\partial z}\right|_{\Gamma_z} = -\frac{D}{2A}(\boldsymbol{m}e_z) \quad (3)$$

are applied in the $z$-direction. $\Gamma_z$ is a boundary of the thin plate of magnet. To calculate the demagnetizing field of the thin film, we used Mumax3 build-in function SetPBC(10,10,0), that is, the demagnetizing field generated by 440 copies of the simulated area was taken into account[33,44]. We used the following parameters: $M_{\text{sat}} = 180$ kA/m, $A = 3.83$ pJ/m, and $D = 0.332$ mJ/m² (ref. [30]). Simultaneously, we varied $K_c$ and $B$ in the range of $-1.75K_0$ to $+1.75K_0$ and $2.25B_d$–$3.25B_d$, respectively. Here, $K_0 = D^2/4A$ and $B_d = D^2/2M_{\text{sat}}A$ are normalization constants. The magnetic field was applied perpendicularly downward to the plane. Magnetic structures are determined by changing the number and configuration of skyrmions in the calculation area and comparing the total energy.

**LTEM image simulation.** LTEM image simulation is based on a Fourier approach[46–50]. The phase of the electron beam ($\varphi$) is given by

$$\varphi(x, y) = -\frac{e}{\hbar}\int A_z(x, y, z)\,\mathrm{d}z, \quad (4)$$

where $\hbar$, $e$, and, $A_z$ are the reduced Planck constant, the elementary electric charge, and the $z$-component of the vector potential. $\varphi$ in reciprocal space is calculated as

$$\tilde{\varphi}(k_x, k_y) = \frac{ie\mu_0 M_{\text{sat}}t}{\hbar}\frac{\tilde{m_x}(k_x, k_y)k_y - \tilde{m_y}(k_x, k_y)k_x}{k_x^2 + k_y^2}, \quad (5)$$

where $\mu_0$ is the permeability of free space. $k_x$ and $k_y$ are the $x$- and $y$-components of the spatial frequency, respectively. $m_x$ and $m_y$ are average values along the thickness direction of $x$- and $y$-components of $\boldsymbol{m}$, respectively. We note that stray field, namely, leakage field and demagnetizing field (see Supplementary Fig. 4b, c), is ignored. LTEM detects magnetization and stray field. The integrated in-plane component value of the stray field is smaller than that of magnetization multiplied by the permeability of free space by one order of magnitude. Thus, LTEM images are simulated by only magnetization. The electron disturbance is calculated by

$$g(k_x, k_y) = \iint \exp\{i\varphi(x, y)\}\exp\left\{-2\pi i\left(k_x x + k_y y\right)\right\}\mathrm{d}x\mathrm{d}y. \quad (6)$$

By considering transfer function

$$t(k_x, k_y) = A(k_x, k_y)\exp\left[-2\pi i\left\{\frac{C_s\lambda^3\left(k_x^2 + k_y^2\right)^2}{4} - \frac{\lambda\Delta f\left(k_x^2 + k_y^2\right)}{2}\right\}\right] \quad (7)$$

and Gaussian distribution

$$E_s(k) = \exp\left\{-\left(\frac{\pi\alpha}{\lambda}\right)^2\left(C_s\lambda^3 k^3 + \Delta f\lambda k\right)^2\right\}, \quad (8)$$

the LTEM image intensity is calculated as

$$I(x, y) = \left|\iint g(k_x, k_y)t(k_x, k_y)E_s(k)\exp\left\{2\pi i\left(k_x x + k_y y\right)\right\}\mathrm{d}k_x\mathrm{d}k_y\right|^2. \quad (9)$$

$A(k_x, k_y)$ is the pupil function and can be assumed to be constant for all reciprocal space. $C_s$, $\lambda$, and $\alpha$ are the spherical aberration of the objective lens, the wavelength of propagating wave, and the beam divergence angle, respectively. We used the following parameters: $\lambda = 2.51$ pm, $t' = 200$ nm, $C_s = 8000$ mm and $\alpha = 10$ µrad (refs. [51,52]).

## Data availability

The data that support the results of this study are available from the corresponding authors upon reasonable request.

## Code availability

The micromagnetic simulation code (Mumax3) is available at https://mumax.github.io/ and the LTEM image simulation code is available from the corresponding authors upon reasonable request.

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

## Acknowledgements
We thank K. Shibata, M. Araidai for helpful discussions and K. Uchida, A. Akama, K. Higuchi, Y. Yamamoto, S. Harada, K. Ishikawa, and H. Cheong for technical support of experiments. This research was financially supported by the Murata Science Foundation and Grant-in-Aid for Scientific Research (C) (JSPS, 18K04679) to Y.-G.S., Grant-In-Aid for Young Scientists (B) (JSPS, 15K17726), Grant-in-Aid for Scientific Research (B) (JSPS, 19H01824) and JST-CREST (JPMJCR16F2) to Y.K., Grant-in-Aid for Scientific Research (C) (JSPS, 18K03529) and Advanced Physical Property Open Unit (APPOU), Hokkaido University to H.K.Y., Grant-in-Aid for Scientific Research (B) (JSPS, 17H02737), Grant-In-Aid for Young Scientists (B) (JSPS, 17K14117), JST-Mirai Program (JPMJMI18G2), Japan, Grant-in-Aid for Scientific Research on Innovative Areas Topological Materials Science (JP15H05853), Grant-in-Aid for Challenging Research (Exploratory) (JSPS, 20K20899), Young Researcher Grant from Center for Integrated Research of Future Electronics, Institute of Materials and Systems for Sustainability, Nagoya University, Nanotechnology Platform Project, MEXT, Japan.

## Author contributions
T.N. and M.N. designed the experiment and wrote the manuscript with input from all authors. Y.G.S. and H.Y. synthesized bulk polycrystalline Co-Zn-Mn. H.K.Y. performed magnetization measurements. T.N. prepared the thin-films, performed LTEM experiments, micromagnetic simulation, and LTEM image simulation. Y.K. made significant contributions to discussion about bimerons. T.I., Y.T., K.S., and N.I. discussed the data and commented on the manuscript. M.K and M.N. supervised the study.

## Competing interests
The authors declare no competing interests.
