## [Peer Review File · Nature Communications]

REVIEWERS' COMMENTS

Reviewer #1 (Remarks to the Author):

Review of Nagase et al. 'Observation of domain wall skyrmions in chiral magnets' transferred to Nature Communications.

The authors have given satisfactory answers to my previous comments and I recommend the paper be published with the changes to the English recommended below:

p2 l38: Change to : 'Domain walls are interesting not only for understanding nature but also for applications.'

p2 l42: Start a new paragraph for the sentence beginning 'A skyrmion ...'

p5 l113: I think 'combination' should be used rather than 'concerted mechanism'.

p6 l125: Should 'compose domains' be 'composite domains'?

Reviewer #2 (Remarks to the Author):

The manuscript "Observation of domain wall bimerons in chiral magnets" has been highly improved as compared to the previously submitted one. The change of denomination, from skyrmions to bimerons, make the description of the texture clearer and easier to understand. Also, the description of the experimental details does not suffer anymore from major lack. In my opinion, this manuscript can be published in the present form and I recommend the publication in Nature Communications.

Reviewer #3 (Remarks to the Author):

Despite of the efforts of the authors, the revised manuscript is less clear and raises more questions than the original one. It is not clear how the authors distinguish between bi-merons and skyrmions, given that each skyrmion ultimately ends at a Bloch point, which is a meron according to the manuscript . Furthermore, the presentation is rather confusing as the introduction, which should also serve as justification of the work is centered around skyrmions, and merons are mentioned just at the very end, like bi-products of skyrmions. The description of the experimental results is also not clearly laid out and the weak english formulation of the previous version of the manuscript has not improved. The points raised in my previous report have not been addressed and the replies to the referee's comments are also not convincing.

Dear Reviewers,

We would like to thank you for the time that has been put into reading our manuscript and providing us with valuable feedback again. We appreciate your comments and believe we have been able to improve the manuscript with your suggestions. The point-by-point responses to your comments below include the location of changes within the manuscript that have been made considering each suggestion.

Reply to Reviewer #1

Review of Nagase et al. 'Observation of domain wall skyrmions in chiral magnets' transferred to Nature Communications.

The authors have given satisfactory answers to my previous comments and I recommend the paper be published with the changes to the English recommended below:

We sincerely appreciate positive comment for our study. Also, your previous comments have improved our manuscript. Thank you again. We have revised the manuscript according to your recommendations as follows.

Change to : 'Domain walls are interesting not only for understanding nature but also for applications.'

We have revised this sentence. Please see lines 39-40 in the main text.

p2 142: Start a new paragraph for the sentence beginning 'A skyrmion ...'

We have revised the sentence as a new paragraph. Please see line 60 in the main text.

p5 1113: I think 'combination' should be used rather than 'concerted mechanism'.

We have changed the words. Please see lines 198, 259, 337, and 352 in the main text.

p6 1125: Should 'compose domains' be 'composite domains'?

Regarding this, the description in the previous revision was ambiguous. The meaning is "Truncated in-plane helical spin orders leading to a finite in-plane net magnetization are the magnetic structure of the domains.". So, we have revised the sentence. Please see line 209 in the main text.

Reply to Reviewer #2

The manuscript "Observation of domain wall bimerons in chiral magnets" has been highly improved as compared to the previously submitted one. The change of denomination, from skyrmions to bimerons, make the description of the texture clearer and easier to understand. Also, the description of the experimental details does not suffer anymore from major lack. In my opinion, this manuscript can be published in the present form and I recommend the publication in Nature Communications.

We sincerely appreciate positive comment for our study. Also, your previous comments were essential. Thanks to your help, the formation process and the academic value of DW bimerons have been clarified and the manuscript has been improved.

Reply to Reviewer #3

Despite of the efforts of the authors, the revised manuscript is less clear and raises more questions than the original one. It is not clear how the authors distinguish between bi-merons and skyrmions, given that each skyrmion ultimately ends at a Bloch point, which is a meron according to the manuscript. Furthermore, the presentation is rather confusing as the introduction, which should also serve as justification of the work is centered around skyrmions, and merons are mentioned just at the very end, like by-products of skyrmions. The description of the experimental results is also not clearly laid out and the weak English formulation of the previous version of the manuscript has not improved. The points raised in my previous report have not been addressed and the replies to the referee's comments are also not convincing.

Previous comments:

The manuscript by Nagase et al., shows new LTEM experimental results obtained on a $\text{Co}_{8.5}\text{Zn}_{7.5}\text{Mn}_4(110)$ film, the thickness of which is about 50 nm, thus about one third of the helical periodicity in this system (~145 nm). By applying a magnetic field perpendicular to the plane of the film they observe the emergence of objects that form chains, alongside domain walls that are oriented along the [1-10] direction. By combining these experimental findings with micromagnetic simulations they come to the conclusion that these objects are skyrmions occurring along domain walls when the magnetic moments are antiparallel to the magnetic field.

The experimental findings are novel and of high quality. The work is an important contribution to the ongoing discussion on the stabilization of skyrmions by domain walls in thin – or even free standing – films.

On the other hand, a main weakness of the manuscript is that it does not go beyond a bare description of the results and does not clarify the important issue of the stability conditions of these domain wall skyrmions. In their previous work (reference 26) the authors performed similar measurements on 100-200 nm thick $\text{Co}_{8.5}\text{Zn}_{7.5}\text{Mn}_4(110)$ films. In these previous measurements they did not see domains or domain walls. Instead, they reported a change in the structure of skyrmions with decreasing thickness: from a regular hexagonal structure at a thickness of 150 nm to a rather disordered and fluctuating one for a thickness of 100 nm (see e.g. the LTEM image collected at 334 K and a field of 160 mT shown in Figure 1b of ref. 26).

A comparison of these previous results with those of the manuscript reveals that there is a dramatic change of behavior when the thickness decreases from 100 nm to 50 nm. In fact Fig. 3 d-e and Fig. 4 imply that for a thickness of 50 nm the domain walls are not between helical/conical domains but between ferromagnetic domains. The origin of this change and the underlying microscopic mechanisms are not addressed in the manuscript, at least not in a clear and convincing way. In this

respect it would also be useful to provide an experimental phase diagram, as done in ref. 26, which obviously would be very different for 50 nm than for 100 nm.

An important question is whether domain-wall skyrmions appear only in very thin samples, with a thickness of about 1/3 of the helical pitch. If this is the case, then it is unclear how these results can contribute to understanding the stabilization mechanisms of low temperature skyrmions in bulk materials like Cu_2OSeO_3 , as argued in the manuscript.

To conclude, despite the interesting results the manuscript leaves several questions open and these should be addressed before it can be considered for publication in Nature Physics.

Composition of the main text and distinction between skyrmions and merons:

We are sorry, it may be not clear how we distinguish between bimerons and skyrmions in the previous revision. Related to this, as you further pointed out, the previous paper structure may confuse the reader because of the sudden appearance of the description of DW bimerons. Therefore, we have revised the manuscript again.

In the previous revision, we mentioned DW bimerons for the first time in the Results section (LTEM observations). But in the present revision, we have mentioned DW bimerons in the Introduction section. Specifically, we have revised the paragraph describing DW skyrmions in the Introduction section, and the name has been unified as DW bimerons from this paragraph based on the different definitions of a skyrmion and a meron. Please see lines 67-83 in the main text.

The revised construction in the paragraph is as follows.

1. Topological defects embedded in or combined with DWs have been known for a long time. For example, Malozemoff and Slonczewski studied Bloch lines as line defects in two-dimensional DWs (ref. 11. *Phys. Rev. Lett.* **29**, 952 (1972)).
2. It is noted that, usually, Bloch lines are formed by accidental excitation.
3. In contrast to accidentally excited Bloch lines, pairs of stabilized Bloch lines in DWs have been predicted by Fal'ko and Iordanskii (ref. 12 in the present revision and ref. 1 in the previous revision). Such topological defects are then called DW skyrmions.
4. However, 'DW skyrmions' is not an accurate name. The reason is as follows.

Mathematically, skyrmions are classified by the second homotopy group $\pi_2(S^2)$, whereas merons are classified by the relative second homotopy group $\pi_2(S^2, S^1)$ (refs. 13,14). The difference is the boundary condition on a circle surrounding the object: the magnetization direction around a skyrmion is fixed, meanwhile that around a meron winds with nonzero winding number $\pi_1(S^1)$. Thus, the topological defect that can exist inside a DW is not a skyrmion but a meron, and we refer to as not DW skyrmions but DW bimerons.

In response to the above revisions, we have changed the word "skyrmions" to "bimerons" and revised the sentences as appropriate after this paragraph, and revised the first sentence of the Abstract section. In addition, DW bimerons predicted in other systems (refs. 17-23) mentioned in this revised paragraph are slightly different from the above definition of topology, so we have revised and described them together as "similar topological defects". Please see line 80 in the main text.

The magnetic flux density map has no certainty as to whether the terminations of the chained bimerons are merons or not. Therefore, we have described a note "The termination of the chains such as in the upper right in Fig. 3a,b,c may not have a topological charge of 1/2 and may have that of zero.", as shown in lines 187-189 in the main text.

Depiction of the magnetic structure of domains in the figures:

We apologize for misleading you about the depiction of the magnetic structure of domains in Fig. 3d,e and Fig. 4. All of the figures show perpendicularly averaged magnetization to the plane of the thin films.

As described in the previous revision, in the thin films, the actual magnetic structures of domains are an in-plane helical spin order under a zero magnetic field and an in-plane conical spin order under a magnetic field, which are truncated before going around once because the film thickness (t) is less than the helical/conical period (λ), as shown in the right figure. Therefore, truncated in-plane helical/conical spin orders result in a finite in-plane net magnetization.

In Fig. 3d,e, the arrows in the domains are oriented in the two $\langle 100 \rangle$ directions of the magnetic easy axes, which depict "perpendicularly averaged truncated in-plane conical spin orders to the plane". In Fig. 4, the colours indicate the orientation and magnitude of "perpendicularly averaged in-plane components of truncated in-plane conical magnetization to the plane".

After you pointed out, we noticed that the main text and captions lack the above explanation and then have added the sentence. Please see lines 247-248, 291-292 in the main text and lines 657-659, 671-672 in the captions.

Formation conditions of DW bimerons and Cu_2OSeO_3 as an example:

We apologize for unclear response to your previous comments and misleading the description of the summarized formation conditions of DW bimerons in the paragraph after the results of micromagnetic simulations. The initial and previous manuscripts gave the impression that a film thickness of 1/3 of the helix/conical period is an important or the threshold for the DW bimeron formation. This description has misled you and will mislead the readers as well. It is important that the helical/conical spin orders are truncated before going around once which results in a finite net magnetization, as shown in the above figure. The correct formation conditions of DW bimerons are summarized as follows:

1. Cubic chiral magnets have large magnetocrystalline anisotropy.
2. The thin films have the (110) plane.
3. The thin-film thickness is not equal to the integral multiple of the helical period for having a finite net magnetization of domains.
4. Dipolar interaction is fully effective to form multidomains.

In the previous revision, the description of the condition (3) was misleading, and effective dipole interaction was missing from the condition. So, we have corrected the former and added the latter. Please see lines 310-312 in the main text.

The reason why Cu_2OSeO_3 is a candidate for the appearance of DW bimerons is that this material is known to have a *large magnetic anisotropy* (refs. 36,37) as well as Co-Zn-Mn. Therefore, $\text{Cu}_2\text{OSeO}_3(110)$ thin-films when the above conditions are satisfied is expected that bimerons are formed as DWs in the vicinity of the Curie temperature of $T_C \sim 60$ K due to thermal fluctuations and bound to DWs in temperatures away from T_C , as well as the present study of the Co-Zn-Mn(110) thin-films described in lines 190-193 in the main text. The previous revision lacked detailed descriptions related to the candidate material Cu_2OSeO_3 , so we have added the sentences in the main text. Please see lines 319-323 in the main text.